# Navigating the Complement Pathway to Optimize PNH Treatment with Pegcetacoplan and Other Currently Approved Complement Inhibitors

**DOI:** 10.3390/ijms25179477

**Published:** 2024-08-31

**Authors:** Peter Hillmen, Regina Horneff, Michael Yeh, Martin Kolev, Pascal Deschatelets

**Affiliations:** 1Apellis Pharmaceuticals, Inc., Waltham, MA 02451, USA; 2Swedish Orphan Biovitrum AB, 171 65 Stockholm, Sweden

**Keywords:** complement, paroxysmal nocturnal hemoglobinuria, C3 inhibitor, C5 inhibitor, factor B inhibitor, factor D inhibitor, pegcetacoplan

## Abstract

Paroxysmal nocturnal hemoglobinuria (PNH) is a rare and potentially life-threatening hematologic disorder caused by a somatic mutation in a relevant portion of hematopoietic stem cells. Mutation of the phosphatidylinositol glycan biosynthesis class A (*PIGA*) gene prevents the expression of cell-surface proteins, including the complement regulatory proteins CD55 and CD59. With decreased or a lack of CD55 and CD59 expression on their membranes, PNH red blood cells become susceptible to complement-mediated hemolysis (symptoms of which include anemia, dysphagia, abdominal pain, and fatigue), leading to thrombosis. State-of-the-art PNH treatments act by inhibiting the dysregulated complement at distinct points in the activation pathway: late at the C5 level (C5 inhibitors, eculizumab, ravulizumab, and crovalimab), centrally at the C3 level (C3/C3b inhibitors and pegcetacoplan), and early at the initiation and amplification of the alternative pathway (factor B inhibitor, iptacopan; factor D inhibitor, danicopan). Through their differing mechanisms of action, these treatments elicit varying profiles of disease control and offer valuable insights into the molecular underpinnings of PNH. This narrative review provides an overview of the mechanisms of action of the six complement inhibitors currently approved for PNH, with a focus on the C3/C3b-targeted therapy, pegcetacoplan.

## 1. Paroxysmal Nocturnal Hemoglobinuria

Paroxysmal nocturnal hemoglobinuria (PNH) is a rare, life-threatening hematologic disorder characterized by severe chronic hemolysis, bone marrow dysfunction, and thrombosis [1,2,3,4,5]. In patients with PNH, an acquired somatic mutation in the phosphatidylinositol glycan biosynthesis class A (*PIGA*) gene occurs in hematopoietic stem cells, resulting in a progeny of glycosylphosphatidylinositol (GPI)-deficient (PNH) blood cells [6]. Because of the *PIGA* mutation, the GPI-anchored surface proteins, including the complement inhibitor proteins CD55 and CD59, are partially (type II) or completely (type III) absent from the surface of PNH cells [6,7]. Without CD55 and CD59, PNH red blood cells are not protected from destruction by the complement system, whereas activated PNH white blood cells and platelets create a favorable environment for thrombosis [7,8,9]. Clinical consequences of PNH include thrombosis, the leading cause of death for untreated patients [8], anemia, smooth-muscle dystonia, and fatigue [1].

PNH is a clonal, nonmalignant disease, in which expansion of PNH cells varies among patients and generally becomes significant in the context of immune-mediated bone marrow failure [10,11]. For example, in acquired aplastic anemia, autoimmune attack on stem cells is less active against GPI-deficient PNH stem cells than against residual normal cells [12,13]. This results in a selective survival advantage for PNH stem cells, which allows them to proliferate and form PNH clones [11,13,14]. Immunological pressure on hematopoiesis has also been suggested to increase the prevalence of PNH-type cells in patients with diffuse large B-cell lymphoma or acute lymphoblastic leukemia within two to three months of receiving CD19-targeted chimeric antigen receptor T-cell therapy [15]. Bone marrow failure syndromes are associated with the development of PNH clones in approximately 60% of patients with aplastic anemia, 15% of those with low-risk myelodysplastic syndromes, and less frequently, in patients with myeloproliferative neoplasms [5,11]. The PNH clone size is determined by flow cytometry analysis of white blood cells—particularly granulocytes and monocytes—rather than red blood cells, because PNH red blood cells are subject to complement-mediated hemolysis [10]. Therefore, the proportion of PNH white blood cells usually represents the PNH clone size, which is related to clinical manifestations [5,11]. However, analysis of both PNH white blood cells and red blood cells is warranted for PNH diagnosis [10]. The complement system is an integral part of the immune system, which offers protection against some bloodborne pathogens and clearance of apoptotic cells [16,17]. The important roles for complement in the control of immunity and homeostasis are evident from the complement dysfunction observed in a range of inflammatory and autoimmune conditions [17,18,19]. The complement system consists of more than 50 component proteins, regulators, and receptors, present in plasma, expressed on cell membranes, and located on or in subcellular compartments [18,19]. As shown in Figure 1, three differentially triggered activation pathways—the classical pathway (CP), the lectin pathway (LP), and the alternative pathway (AP)—converge to the central complement component C3 and lead to a common cytolytic pathway [18,19] and activation of complement receptors. The cytolytic pathway is initiated by the cleavage of C5 and subsequent assembly of the lytic membrane attack complex (MAC; C5b-9) in targeted pathogens and cells [16,17,18]. Concomitant with the cytolytic pathway activation, the intermediary cleavage fragments of C3 and C5 enhance the inflammatory response (via anaphylatoxins, C3a and C5a) and opsonize pathogens and damaged or apoptotic cells (via opsonins, C3b and subsequent fragments) [17,18,19]. Anaphylatoxins C3a and C5a bind to specific G-protein-coupled receptors (C3a receptor, C5a receptor 1, and C5a receptor 2) to mediate a range of outcomes, including inflammation [18]. Anaphylatoxins can also act locally and intracellularly to regulate the immune system and affect cell survival [20].

The system is designed to react rapidly and mount a powerful response to eliminate threatening targets. Pathogen surface-bound antibody complexes trigger the CP, and bacterial surface polysaccharides initiate the LP. In contrast, low levels of spontaneous and continuous C3 hydrolysis at an internal thioester bond that can covalently attach to injured surfaces drive activation of the AP in a process known as tick-over. Upon activation, enzymatic complexes, called C3 convertases, are formed (C4bC2a via the CP and LP; C3bBb via the AP) and cleave C3 into the anaphylatoxin C3a and the opsonin C3b. Deposition of C3b and subsequent fragments, such as inactivated C3b (iC3b), C3c, C3dg, and C3d, on cell membranes mobilizes constituents of the AP (i.e., factor D, which cleaves factor B into the active fragment Bb) to assemble new C3 convertases (C3bBb) and drives a self-amplifying opsonization cycle (amplification loop) that produces large amounts of C3b [19,22]. The AP amplification loop provides a powerful way to strengthen complement activation at targeted surfaces after initiation by any of the three activation pathways. Several regulatory mechanisms are in place to maintain a focused and controlled response, including early inhibition of the CP or LP (e.g., serpin C1 inhibitor), prevention of C3b and C4b deposition on host cells (e.g., factor I and cell-bound cofactors, such as factor H and CD46), limitation of C3 and C5 convertase assembly and activity (e.g., complement receptor 1 (CR1), factor H, factor I, CD46, and CD55), and inhibition of MAC assembly (e.g., CD59) [17,18,19].

Clinical features, laboratory markers, and flow cytometric determination of PNH clone size are used to determine PNH subcategories, which then inform treatment recommendations from the International PNH Interest Group (IPIG) [7,10]: Patients with classical PNH have clinical manifestations of hemolysis or thrombosis, PNH clone size > 50%, and bone marrow with erythroid hyperplasia. Patients with PNH in the setting of another bone marrow syndrome, such as aplastic anemia or myelodysplastic syndrome, have mild anemia without clear laboratory signs of hemolysis, and variable but usually small PNH clone size (<50%). The final subset of patients has subclinical PNH, with a low PNH clone size (<10%) and no clinical or laboratory evidence of hemolysis [7,10]. Laboratory markers include increased lactate dehydrogenase (LDH) concentrations, decreased hemoglobin concentrations, and increased absolute reticulocyte counts (ARCs) in patients with classical PNH who have significant clinical symptoms and do not have concomitant bone marrow failure (Table 1) [9,23]. Complement inhibitors are generally indicated for patients with classical PNH, whereas the degree of hemolysis contribution to anemia determines the need for complement inhibition in patients with PNH associated with bone marrow failure. Patients with asymptomatic PNH usually do not require treatment with complement inhibitors [3,5,10]. Complement inhibitors are also recommended during pregnancy, because complement activation increases, especially after 20 weeks of gestation. Thromboembolic events were the main cause of maternal mortality, while premature delivery was the main cause of fetal mortality [3]. Eculizumab therapy was shown to decrease maternal complications and improve fetal survival in an International PNH registry study of 75 pregnancies in 61 women with PNH [24]. In addition, the first successful pregnancy during pegcetacoplan treatment has been reported in a woman with PNH who had two prior miscarriages while receiving eculizumab [25].

In recent years, PNH has emerged as the prototypical complement-mediated disease because its pathogenesis is likely to be almost entirely driven by dysregulation of the AP of complement activation on the surface of PNH red blood cells [5,22]. Insights gained from PNH research are likely to apply to other diseases/disorders driven by complement dysregulation, including geographic atrophy secondary to age-related macular degeneration and the rare kidney diseases C3 glomerulopathy (C3G) and immune complex membranoproliferative glomerulonephritis (IC-MPGN) [18,19,31,32,33]. However, with most complement-mediated diseases, it is difficult to isolate the role of complement as clearly as in PNH due to additional underlying etiologies. Furthermore, the potential clinical trial endpoints are more rapidly reached and better defined in PNH. Consequently, PNH has emerged as an ideal indication for the testing of multiple complement inhibitors, leading to a greater choice of treatments for patients with classical PNH, according to IPIG treatment recommendations [5] (Table 2 and Figure 2). The following sections provide a comparative overview of pegcetacoplan, an inhibitor of C3 and C3b, and other currently approved complement inhibitors for PNH.

## 2. Eculizumab, Ravulizumab, and Crovalimab, Terminal Complement Inhibitors for PNH

Therapeutic advances have changed PNH treatment from supportive care for symptom reduction [8,45] to complement-directed treatment for disease control [5,46]. The first complement-directed therapies for PNH inhibited the terminal cytolytic portion of the complement cascade by binding and inhibiting C5 [47]. Eculizumab was the first C5 inhibitor for PNH, approved in 2007, with maintenance dosing administered intravenously (IV) every two weeks [34,35]. More than 10 years later, ravulizumab was the next C5 inhibitor approved for patients with PNH [36,37]. Ravulizumab had a similar mechanism of action to eculizumab but with a longer half-life, allowing for IV maintenance dosing every eight weeks, and an efficacy that was noninferior to eculizumab [36,48,49]. A third C5 inhibitor, crovalimab, was approved by the US Food and Drug Administration (FDA) in 2024 for adult and pediatric patients 13 years and older with PNH [38]. After IV administration of loading doses, crovalimab is administered every four weeks by subcutaneous injection [38,39,40,41,42,43,44,45,46,47,48,49,50,51]. Eculizumab, ravulizumab, and crovalimab bind C5 to prevent C5b generation and MAC assembly, allowing PNH red blood cells to avoid complement-mediated intravascular hemolysis (IVH) [45,48,49,50,51,52] (Figure 3).

The clinical improvements with C5 inhibitors in patients with PNH were marked; concentrations of LDH, a marker of IVH, decreased dramatically, indicating a reduction of IVH [45,49,50,51,52,53]. In addition, the clone size of PNH red blood cells usually increased, and the clone size of PNH white blood cells did not change, concentrations of hemoglobin stabilized most often at levels below normal, and ARC remained above normal [45,49,50,51,52,53]. Patients also had decreased transfusion needs with eculizumab, ravulizumab, and crovalimab [45,50,51,52]. Subsequent analysis of long-term data from more than 4000 patients in the International PNH Registry showed that eculizumab treatment led to a 49% reduction in mortality [27]. Likewise, long-term data from 509 UK patients with PNH showed that ravulizumab- or eculizumab-treated patients who did not require treatment for bone marrow failure had overall survival rates comparable to those of age- and sex-matched controls [23]. 

Despite these marked gains with eculizumab and ravulizumab, it became evident that C5 inhibition for PNH had inherent limitations [14]. Although the terminal portion of the complement cascade is blocked, proximal C3 activation of the cascade continues unchecked [30,54]. This causes an excessive C3b deposition on the surface of PNH red blood cells, which lack the complement regulator CD55-negative and do not express CD46. Although C5 inhibition blocks MAC-mediated lysis, the proximal complement cascade continues to be active. After PNH red blood cells reach a threshold of C3b deposition, they are engulfed by phagocytes (via interaction of C3dg fragments with complement receptor 3 (CR3)) in the reticuloendothelial system of the liver and spleen, in what is designated as extravascular hemolysis (EVH) [30,47,54,55]. This hemolytic activity can be initiated by both the chronic, low-grade activation of the AP and complement-amplifying conditions (CACs, e.g., infections, vaccination, and tissue injury) that activate the CP and LP [56]. The ongoing EVH with C5 inhibitors can be observed in most patients as persistent, mildly elevated LDH concentrations, undetectable haptoglobin, elevated bilirubin concentrations, persistent reticulocytosis (i.e., ARC above normal), and chronic anemia (i.e., hemoglobin concentrations that were stabilized but lower than normal) [30,54]. Consequences of this chronic anemia include dyspnea, fatigue, and residual transfusion needs [23,57].

A suboptimal response to complement inhibition or a CAC can predispose patients with PNH to breakthrough hemolysis (BTH) events [53,58,59]. During BTH events, patients experience a resurgence of IVH and the return of PNH symptoms due to incomplete disease control [59,60]. Long-term studies demonstrated that between 11% and 27% of patients who received approved dosages of eculizumab for PNH reported a BTH event [53,61]. More recent findings indicate that rates of BTH events during complement inhibitor treatment for PNH are even higher, with 75% of patients experiencing BTH during a median of six years of follow-up. These findings differ from earlier studies in that they include patients who received eculizumab and eventually switched to a different complement inhibitor [62]. Most BTH events are caused by either inadequate C5 inhibition (i.e., pharmacokinetic events) or CACs that increase complement activity to a degree that overpowers the inhibitor’s complement control (i.e., pharmacodynamic events) [45,49]. Noninferiority trials of ravulizumab versus eculizumab in patients with PNH demonstrated that BTH events also occurred in ravulizumab-treated patients, either associated with CACs or without a known cause [48,49,60].

## 3. Pegcetacoplan, the First C3 Complement Inhibitor for PNH

Pegcetacoplan, the next therapy for patients with PNH, was approved in 2021 [39,40]. Pegcetacoplan, administered via subcutaneous infusion twice a week [39,40], addressed the challenge of chronic anemia by providing more complete complement cascade inhibition [63]. This broad inhibition is critical to treat the uncontrolled AP complement activation on PNH red blood cell membranes that is observed in PNH [63]. As discussed earlier, C3 cleavage into C3a and C3b is the convergence point for all three complement activation pathways [18,19]. Then, the amplification loop starts with C3b generation (by either CP, LP, or AP) and deposition, and the additional C3b amplifies the strength of complement activation via the AP amplification loop [19,22]. Various complement regulators (e.g., CR1 and factor I for red blood cells) and proteases act on C3b fragments to generate iC3b and further breakdown fragments, such as C3c, C3dg, and C3d (Figure 4) [18].

Pegcetacoplan was developed as a derivative of compstatin, a cyclic peptide that binds to C3, C3b, and C3c [64,65]. Pegcetacoplan incorporates two compstatin analogs linked with a polyethylene glycol molecule to increase its half-life and solubility [18,66]. With an affinity for C3 and C3b (as well as iC3b and C3c), similar to that of compstatin [64], pegcetacoplan provides complement blockade for various indications by directly inhibiting the cleavability of C3 by the two C3 convertases (C4bC2a from the CP/LP and C3bBb from the AP) [18,64,67]. Pegcetacoplan also inhibits the convertases by blocking C3b: the C3 convertase of the AP (C3bBb) and the C5 convertases of the AP (C3bC3bBb) and CP/LP (C4bC2aC3b) [18,64,67]. Pegcetacoplan specifically suppresses the AP at three different points (the tick-over initiation mechanism, the AP C3 convertase, and the AP C5 convertase), which reduces chronic low-grade complement activation [18,68]. Thus, pegcetacoplan provides broad complement inhibition by blocking all three proximal complement activation pathways at the C3 level and suppressing the amplification loop of the AP [63,64,65]. The result is more complete proximal (through C3 and AP C3 convertase inhibition) and terminal (through both AP and CP/LP C5 convertases’ inhibition) complement cascade control [22,54,59]. This provides a broad control of hemolysis by addressing both IVH and EVH [54,55,59] (Figure 5).

The decreased C3 deposition on PNH red blood cells with pegcetacoplan treatment has been observed in two phase 3 clinical trials. In eculizumab-experienced patients with PNH in the PEGASUS trial, pegcetacoplan decreased C3 deposition when all patients received both pegcetacoplan and eculizumab during the run-in period [69]. During the subsequent randomized control period, patients who received pegcetacoplan monotherapy continued to have low levels of C3 deposition on PNH red blood cells; in contrast, C3 deposition increased in patients who returned to eculizumab monotherapy. Later in the trial, patients in the eculizumab group had reduced C3 deposition on red blood cells when they received add-on pegcetacoplan during the run-in for the open-label period. These levels remained low when they switched to open-label pegcetacoplan monotherapy. In the PRINCE trial, complement-inhibitor-naive patients with PNH had decreased C3 deposition on red blood cells as early as four weeks after beginning pegcetacoplan treatment (data on file). A similar effect of pegcetacoplan on deposition of C3b fragments on relevant cells was described in a clinical trial of patients with the complement-mediated disorder of renal glomerulopathy C3G, where deposition of C3c fragments on renal glomeruli membranes was reduced at 12 and 52 weeks of pegcetacoplan treatment [70].

Another molecular effect of pegcetacoplan is decreased formation of the MAC (i.e., C5b-9) because the C5 convertases (C4bC2aC3b from the CP/LP and C3bC3bBb from the AP) are not generated, thereby resulting in both proximal and terminal complement inhibition [59]. Although C5b-9 levels were not evaluated in pegcetacoplan clinical trials, this was demonstrated in a clinical trial conducted in patients with another complement regulatory disorder, C3G, in which serum C3 concentrations increased and plasma-soluble C5b-9 (sC5b-9) concentrations decreased after 48 weeks of pegcetacoplan treatment [32]. Similarly, in clinical trials of patients with PNH, pegcetacoplan treatment induced a sustained increase in C3 concentrations [71], which could potentially result from a reduced C3 consumption due to the inhibition of complement activation by pegcetacoplan, or from an increased C3 half-life.

The molecular effects of pegcetacoplan translate to clinical benefits. After 16 weeks of pegcetacoplan monotherapy in the PEGASUS trial, patients with PNH who had chronic anemia with a C5 inhibitor and switched to pegcetacoplan had higher hemoglobin concentrations, lower ARC, less fatigue, and fewer transfusions than did patients who continued eculizumab [72,73]. Similar clinical benefits with pegcetacoplan were observed in complement-inhibitor-naive patients with PNH [71,74].

The broad complement cascade inhibition of pegcetacoplan also increases the percentage of PNH red blood cells because PNH red blood cells are further protected from all-cause hemolysis. The percentage of PNH granulocytes does not change because these cells have a normal lifespan in patients with PNH [11,59]. In eculizumab-experienced patients from PEGASUS, the mean percentage of PNH red blood cells increased from 66.8% at baseline to 93.9% after 16 weeks of pegcetacoplan monotherapy, whereas the percentage of PNH red blood cells decreased somewhat in patients who continued eculizumab (72.9% at baseline to 62.6% at week 16) [72]. The percentages of PNH-negative granulocytes were unchanged in both groups through week 16 (data on file). This improved survival of PNH red blood cells with pegcetacoplan demonstrates a reduction in the C3d-mediated EVH in the liver and spleen that had limited PNH red blood cell survival during eculizumab treatment. These findings were maintained in the PEGASUS open-label period [69]. Similar findings were observed in trials of pegcetacoplan in complement-inhibitor-naive patients with PNH [71] (data on file). This demonstrates that pegcetacoplan provided protection from IVH and EVH in patients who had not received a complement inhibitor [74]. Pegcetacoplan’s broad inhibition of the complement cascade has brought with it concerns about the potential for increased risk of serious infections stemming from inadequate complement-mediated immunity [75,76]. This possibility, theoretically inherent to all complement inhibitors due to their mechanism of action, has been addressed by label-required vaccinations and prophylactic antibiotic use when complement inhibitor treatment must be initiated before vaccinations can be confirmed or administered [39,40,59,76]. Furthermore, in conditions of C3 inhibition similar to those obtained with compstatin derivatives and pegcetacoplan, it has been observed that the low residual levels of C3 produced in local tissues can provide opsonic functions adequate for defense against infection [63,77,78]. Another factor to consider is that infection risk will be highly individualized, as each patient has inherited a unique set of complement polymorphisms, known as the complotype, that greatly affect complement activity and infection susceptibility [79]. The absence of encapsulated meningococcal infections among all patients with PNH treated with pegcetacoplan (1127 patient-years of exposure through 13 November 2023) provided real-world support that pegcetacoplan’s labeling guidelines may reduce concerns of an increased risk of infections for pegcetacoplan-treated patients, although continued follow-up and monitoring are needed [80].

The high proportion of PNH red blood cells in pegcetacoplan-treated patients is a result of better control of hemolysis compared to C5 inhibitors but raised the theoretical possibility that BTH events in these patients could be more severe and difficult to manage [59]. Continued studies have shown that pegcetacoplan-treated patients, similar to all complement-inhibitor-treated patients with PNH, are susceptible to BTH events, particularly in the presence of CACs [72,73,74,81]. In a recent integrated analysis of pegcetacoplan phase 3 clinical trials and their open-label extension, BTH events were successfully managed with pegcetacoplan, and patients with these events recovered [81,82]. Approximately one-third of BTH events were associated with a potential CAC [82]. BTH events are known to increase the risk of thrombotic events, which can potentially be an ultimate clinical manifestation of BTH [83]. A recent report showed that the rate of thrombotic events in pegcetacoplan-treated patients was comparable to that of patients receiving C5 inhibitors [84].

The rare kidney disease C3G is another condition characterized by dysregulation of the complement. In C3G, hyperactivation of the complement AP results in the accumulation of C3 and C5 breakdown products in the glomeruli, leading to inflammation and structural kidney damage [85]. Unlike PNH, C3G represents a highly heterogeneous spectrum of conditions that may be due to genetic abnormalities in various complement components (such as C3-, factor B-, factor H-, factor I-, and factor H-related proteins), the development of autoantibodies against complement components, or nephritic factors that stabilize C3 and/or C5 convertases [33]. Nevertheless, complement inhibitors that can target the AP are currently being investigated for C3G, including pegcetacoplan [32,33,70]. In a phase 2 open-label study, pegcetacoplan treatment modulated complement hyperactivity (increasing plasma C3 and decreasing sC5b-9 concentrations) and improved some C3G clinical signs [32]. Early findings from another phase 2, open-label, randomized study in patients with post-transplant recurrence of C3G suggest that pegcetacoplan treatment can reduce C3 deposition in the glomeruli of affected kidneys [71].

## 4. Iptacopan and Danicopan, New Proximal Complement Inhibitors for PNH

Iptacopan, which received approval by the FDA in 2023 and the European Medicines Agency (EMA) in 2024, is the first oral PNH monotherapy (taken two times per day) [41,42]. Iptacopan is a small molecule that was identified for its inhibitory activity of the factor-B-dependent C3 convertase C3Bb of the AP [86]. By binding to factor B and the C3 convertase of the AP (C3Bb) on the membrane of red blood cells, iptacopan also blocks the AP C5 convertase (C3bC3bBb) and controls both C3b-mediated EVH and C5-mediated IVH initiated through the AP C3 convertase (Figure 6) [86,87]. Using in vitro assays and murine models, iptacopan was also shown to inhibit the amplification loop triggered by C3b generated from the CP and LP, although it did not completely inhibit the CP/LP-dependent complement activation [86]. Various in vitro assays have indicated that the CP/LP pathways will not account for more than 50% of total complement activity and that iptacopan does not inhibit pure CP/LP activation, unlike pegcetacoplan [86,88,89]. Danicopan, approved by the US FDA and the EMA in 2024, is the first oral add-on therapy (taken three times per day) to eculizumab or ravulizumab for patients with PNH and EVH receiving a C5 inhibitor [43,44]. Danicopan acts by reversibly binding to factor D [43,44]. As such, danicopan is expected to prevent the formation of Bb; however, contrary to iptacopan, danicopan is not expected to directly inhibit the AP C3 and C5 convertases (C3bBb and C3bC3bBb, respectively) once they are formed.

The approved dose of iptacopan was shown to inhibit AP activity (measured by the Wieslab assay) by more than 80% in healthy volunteers, demonstrating limited residual AP hemolysis with factor B inhibition [90]. Factor D inhibition with danicopan monotherapy also led to AP inhibition, although the blockade was not complete in all patients [91].

Clinical trial results have shown that iptacopan, similar to pegcetacoplan, effectively prevented IVH and EVH in patients with PNH who had chronic hemolysis with a C5 inhibitor, as shown by decreased LDH concentrations, increased hemoglobin concentrations, and reduced ARC [87]. After 24 weeks of iptacopan monotherapy, mean C3 deposition on red blood cells had decreased to <1% in eculizumab-experienced patients with PNH, and C3 deposition did not occur in patients who were eculizumab-naive [87]. In both patient populations, the proportion of PNH red blood cells increased to approximately 90% during iptacopan treatment [87].

In clinical trials, danicopan in combination with eculizumab or ravulizumab provided more effective PNH control than a C5 inhibitor alone, with the combination therapy leading to decreased LDH concentrations, increased but below-normal hemoglobin concentrations, and reduced ARCs [92]. However, in eculizumab-treated patients with PNH, C3 deposition only decreased, from 29% at baseline to 13% after 12 weeks of danicopan add-on treatment. The proportion of PNH red blood cells increased by 25% in patients who received add-on danicopan and decreased by 3% without add-on danicopan [92]. 

In two phase 3 clinical trials, the two BTH events in patients receiving iptacopan were mild or moderate [87]. No meningococcal infections occurred in iptacopan-treated patients, one patient had a urinary tract infection caused by an encapsulated bacterium (*Pseudomonas aeruginosa*), and one patient had a serious adverse event of bacterial pneumonia (no causative organism identified) during iptacopan treatment [87]. In the phase 3 trial of add-on danicopan, mild or moderate hemolysis was reported in two patients receiving danicopan, and no meningococcal infections were reported [92].

## 5. Conclusions

The therapeutic landscape for PNH has significantly evolved, with an increasing number of complement inhibitors now available, including the C5 inhibitors eculizumab, ravulizumab, and crovalimab, the C3/C3b inhibitor pegcetacoplan, and the AP complement inhibitors iptacopan and danicopan. Treatment with C5 inhibitors markedly improves clinical signs of PNH and reduces mortality, but does not prevent EVH, which may result in a suboptimal response in most patients. Pegcetacoplan was the first proximal complement inhibitor approved for PNH; by blocking C3 and C3b, pegcetacoplan prevents both IVH and EVH, and has demonstrated long-term safety and treatment benefits in patients with PNH. Iptacopan monotherapy and danicopan add-on therapy to C5 inhibitors are innovative treatments that prevent IVH and EVH initiated through the AP. Although all proximal inhibitors offer added clinical benefits in patients with persistent anemia previously treated with C5 inhibitors, pegcetacoplan is the only complement-targeted therapy approved for PNH that inhibits complement cascade activity proximally after initiation through the CP/LP and AP to provide extensive complement regulation. Demonstration of effective treatments for breakthrough hemolysis events has tempered early concerns about severe breakthrough hemolysis events in patients treated with pegcetacoplan. Among the increasingly diverse and effective options for PNH, pegcetacoplan’s broad pharmacology provides a unique form of complement cascade inhibition to improve outcomes for patients with PNH and potentially with other complement-mediated diseases, such as C3 glomerulopathies.

## Figures and Tables

**Figure 1 ijms-25-09477-f001:**
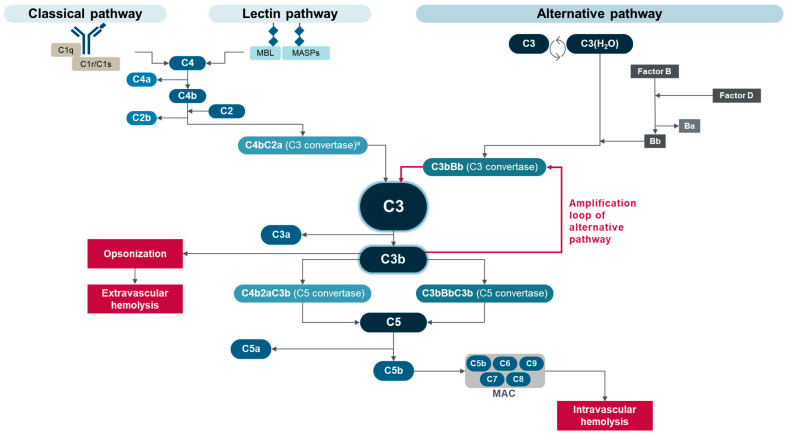
Complement activation in untreated PNH. ^a^ Also referred to as C4bC2b C3 convertase. MAC, membrane attack complex; MASPs, mannose-binding lectin-associated proteases; MBL, mannose-binding lectin. Adapted from Goldberg RA et al., “The evolving treatment paradigm in geographic atrophy: A panel of experts translates clinical trial data to real-world practice”, Retina Today. May/June 2024, ©2024 Apellis Pharmaceuticals, Inc. All rights reserved [21].

**Figure 2 ijms-25-09477-f002:**
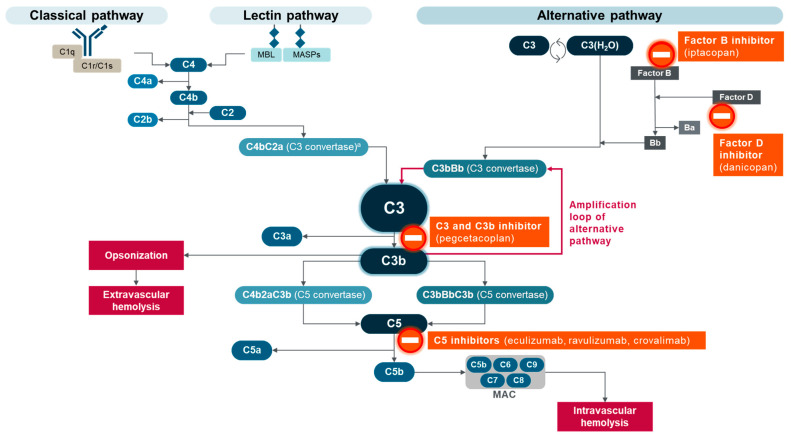
Complement inhibitor targets in PNH. ^a^ Also referred to as C4bC2b C3 convertase. MAC, membrane attack complex; MASPs, mannose-binding lectin-associated proteases; MBL, mannose-binding lectin. Adapted from Goldberg RA et al., “The evolving treatment paradigm in geographic atrophy: A panel of experts translates clinical trial data to real-world practice”, Retina Today. May/June 2024, ©2024 Apellis Pharmaceuticals, Inc. All rights reserved [21].

**Figure 3 ijms-25-09477-f003:**
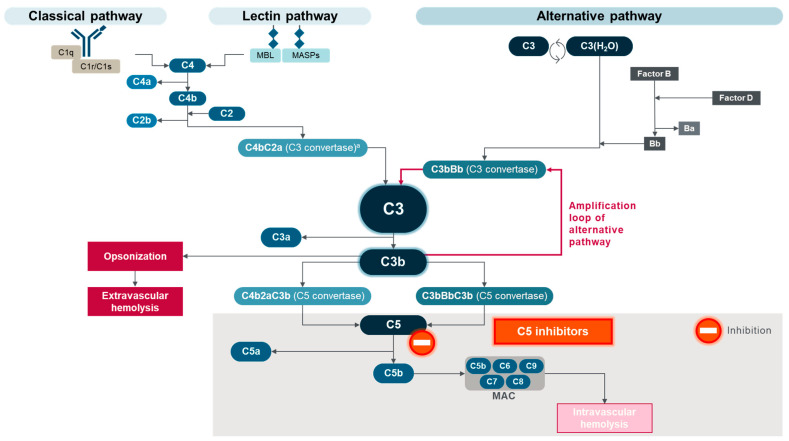
Mechanism of action of C5 inhibitors (eculizumab, ravulizumab, and crovalimab) in PNH. Gray shading indicates portions of complement activation targeted by C5 inhibitors. ^a^ Also referred to as C4bC2b C3 convertase. MAC, membrane attack complex; MASPs, mannose-binding lectin-associated proteases; MBL, mannose-binding lectin; PNH, paroxysmal nocturnal hemoglobinuria. Adapted from Goldberg RA et al., “The evolving treatment paradigm in geographic atrophy: A panel of experts translates clinical trial data to real-world practice”, Retina Today. May/June 2024, ©2024 Apellis Pharmaceuticals, Inc. All rights reserved [21].

**Figure 4 ijms-25-09477-f004:**
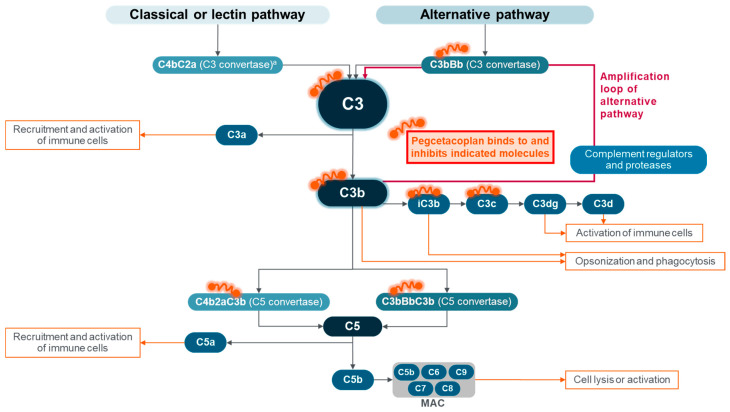
Pegcetacoplan-binding targets. ^a^ Also referred to as C4bC2b C3 convertase. iC3b, inactivated C3b; MAC, membrane attack complex. Adapted from Goldberg RA et al., “The evolving treatment paradigm in geographic atrophy: A panel of experts translates clinical trial data to real-world practice”, Retina Today. May/June 2024, ©2024 Apellis Pharmaceuticals, Inc. All rights reserved [21].

**Figure 5 ijms-25-09477-f005:**
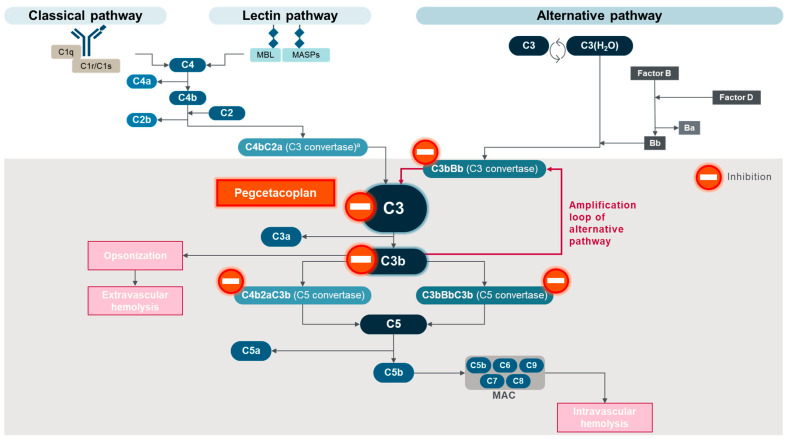
Mechanism of action of the C3 inhibitor pegcetacoplan in PNH. Gray shading indicates portions of complement activation targeted by pegcetacoplan. ^a^ Also referred to as C4bC2b C3 convertase. MAC, membrane attack complex; MASPs, mannose-binding lectin-associated proteases; MBL, mannose-binding lectin; PNH, paroxysmal nocturnal hemoglobinuria. Adapted from Goldberg RA et al., “The evolving treatment paradigm in geographic atrophy: A panel of experts translates clinical trial data to real-world practice”, Retina Today. May/June 2024, ©2024 Apellis Pharmaceuticals, Inc. All rights reserved [21].

**Figure 6 ijms-25-09477-f006:**
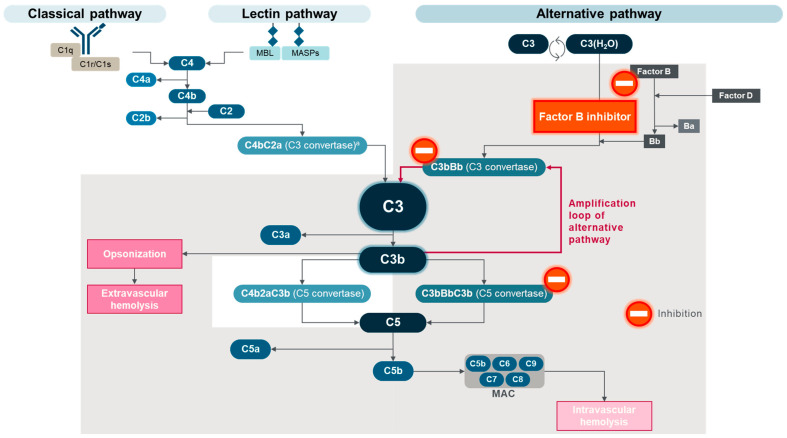
Mechanisms of action of factor B (iptacopan) in PNH. Gray shading indicates portions of complement activation targeted by factor B inhibitor. ^a^ Also referred to as C4bC2b C3 convertase. MAC, membrane attack complex; MASPs, mannose-binding lectin-associated proteases; MBL, mannose-binding lectin; PNH, paroxysmal nocturnal hemoglobinuria. Adapted from Goldberg RA et al., “The evolving treatment paradigm in geographic atrophy: A panel of experts translates clinical trial data to real-world practice”, Retina Today. May/June 2024, ©2024 Apellis Pharmaceuticals, Inc. All rights reserved [21].

**Table 1 ijms-25-09477-t001:** Markers of PNH in untreated patients [9,26,27,28,29,30].

Parameter	Levels in Untreated Patients with PNH	Physiological Cause
LDH or LDH ratio (LDH/ULN) (marker of hemolysis)	High often ≥ 1.5 × ULN (up to 10 × ULN)	Increased due to hemolysis of PNH red blood cells (mostly through IVH)
Hemoglobin (marker of anemia)	Low (≤10 g/dL)	Decreased due to hemolysis of PNH red blood cells (through IVH and EVH) or bone marrow failure
ARC (marker of bone marrow compensation) ^a^	High often > ULN (up to 1.5 × ULN)	Increased due to bone marrow compensation for hemolyzed PNH red blood cells
PNH red blood cell clone size	Variable (usually > 5%)	Increased due to *PIGA* gene somatic mutation and subsequent loss of GPI-linked proteins (e.g., CD59 for red blood cells)
PNH granulocyte clone size (diagnostic marker of PNH)	Variable (usually ≥ 10%)	

^a^ Except in cases with concomitant bone marrow failure. ARC, absolute reticulocyte count; EVH, extravascular hemolysis; GPI, glycophosphatidylinositol; IVH, intravascular hemolysis; LDH, lactate dehydrogenase; *PIGA*, phosphatidylinositol glycan biosynthesis class A; PNH, paroxysmal nocturnal hemoglobinuria; ULN, upper limit of normal.

**Table 2 ijms-25-09477-t002:** Approved complement inhibitors for PNH [34,35,36,37,38,39,40,41,42,43,44].

Inhibition Target	Drug Name	Approval Date	Indication	Administration Regimen	Molecule	MOA
C5	Eculizumab [34,35]	03/2007	Adults (≥18 years) with PNH	IV infusion every 2 weeks	C5 mAB	Inhibits C5 cleavage to C5a and C5b
C5	Ravulizumab [36,37]	12/2018	Adults and children (≥1 month) with PNH	IV infusions every 4 or 8 weeks based on body weight	C5 hmAB	Inhibits C5 cleavage to C5a and C5b
C5	Crovalimab [38]	06/2024	Adults and children (≥13 years) with PNH and body weight ≥ 40 kg	SC injection every 4 weeks	C5 hmAB	Inhibits C5 cleavage to C5a and C5b
C3 and C3b	Pegcetacoplan [39,40]	03/2021	Adults with PNH	SC infusion twice weekly	Pegylated compstatin	Regulates C3 cleavage and generation of downstream effectors
Factor B	Iptacopan [41,42]	12/2023	Adults with PNH	Orally twice daily	Small peptide	Regulates AP-dependent C3 cleavage and AP amplification loop
Factor D	Danicopan [43,44]	03/2024	Add-on to ravulizumab or eculizumab for EVH in adults with PNH	Orally 3 times daily	Small peptide	Prevents factor B cleavage to Ba and Bb, regulating C3 cleavage and AP amplification loop

AP, alternative pathway of complement activation; EVH, extravascular hemolysis; hmAB, humanized monoclonal antibody; IV, intravenous; mAB, monoclonal antibody; MOA, mechanism of action; PNH, paroxysmal nocturnal hemoglobinuria; SC, subcutaneous.

## Data Availability

Data were derived from public domain resources.

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
