# Peer review of "Navigating the Complement Pathway to Optimize PNH Treatment with Pegcetacoplan and Other Currently Approved Complement Inhibitors"

_ijms, 2024, doi:10.3390/ijms25179477_

Round 1

Reviewer 1 Report

Comments and Suggestions for Authors

The authors describe in detail the complement dysregulation in PNH. The description is complete and help to understand better the mechanism of action of the treatment. The authors report all available treatments  approved for PNH. Figures facilitate the overall understanding. The main part of the manuscript focused on Pegcetacoplan, its mechanism of action, and its benefit versus other treatments.

The manuscript could benefit from additional suggestions: 

-Consider making a summary table on all treatments described and one illustration of the three main pathways of complement activation and an overview of some of the targets of therapeutic complement inhibition.

-Figures 3 and 4 are a little redundant. One will be sufficient.

-It will be good to briefly mention the management recommendations of the International PNH Interest Group and particularly on the issue of pregnancy.

Reviewer 2 Report

Comments and Suggestions for Authors

In the manuscript "Naviagting  the Complement Pathway to Optimize PNH Treatment with Pegcetacoplan and other Currently Approved Complement Inhibitors", a team led by Hillmen, describee an overreivew of the mechanism of action of the 6 complement inhibitors currently approved for PNH.

This manuscript presents several interesting points in a well-structured manner, but there are also some minor weaknesses that need to be addressed. Here are the specific points:

1. More details about PIGA mutation need to be provided, such as the percentage of PIGA mutation in patients and the time when the PIGA mutation occurred. Referencing PMID: 34889408 might be helpful.

2. References are needed for line 134 to line 136.

3. It's important to note that a high prevalence of PNH can also be detected in patients undergoing CAR T treatment. Considering PMID: 34396050 for inclusion in this manuscript could be beneficial.
